# The tumor suppressor PTEN and the PDK1 kinase regulate formation of the columnar neural epithelium

Joaquim Grego-Bessa[1†], Joshua Bloomekatz[1‡], Pau Castel[2], Tatiana Omelchenko[3], José Baselga[2,4], Kathryn V Anderson[1*]

[1]Developmental Biology Program, Sloan Kettering Institute, Memorial Sloan Kettering Cancer Center, New York, United States; [2]Human Oncology and Pathogenesis Program, Sloan Kettering Institute, Memorial Sloan Kettering Cancer Center, New York, United States; [3]Cell Biology Program, Sloan Kettering Institute, Memorial Sloan Kettering Cancer Center, New York, United States; [4]Department of Medicine, Memorial Sloan Kettering Cancer Center, New York, United States

*For correspondence: k-anderson@ski.mskcc.org

Present address: [†]Cancer Epigenetics and Biology Program, Bellvitge Biomedical Research Institute, Barcelona, Spain; [‡]Division of Biological Sciences, University of California, San Diego, San Diego, United States

Competing interests: The authors declare that no competing interests exist.

**Abstract** Epithelial morphogenesis and stability are essential for normal development and organ homeostasis. The mouse neural plate is a cuboidal epithelium that remodels into a columnar pseudostratified epithelium over the course of 24 hr. Here we show that the transition to a columnar epithelium fails in mutant embryos that lack the tumor suppressor PTEN, although proliferation, patterning and apical-basal polarity markers are normal in the mutants. The *Pten* phenotype is mimicked by constitutive activation of PI3 kinase and is rescued by the removal of PDK1 (PDPK1), but does not depend on the downstream kinases AKT and mTORC1. High resolution imaging shows that PTEN is required for stabilization of planar cell packing in the neural plate and for the formation of stable apical-basal microtubule arrays. The data suggest that appropriate levels of membrane-associated PDPK1 are required for stabilization of apical junctions, which promotes cell elongation, during epithelial morphogenesis.

## Introduction

Phosphoinositides are powerful second messengers in signaling pathways that also control epithelial organization and cell motility, placing them at a unique intersection of signaling and morphogenesis. The lipid phosphatase PTEN, which converts the membrane lipid phosphatidylinositol (3,4,5)-trisphosphate (PtdIns(3,4,5)P$_3$) to phosphatidylinositol 4,5-bisphosphate (PtdIns(4,5)P$_2$), is the second most commonly mutated gene in human cancers. PtdIns(3,4,5)P$_3$ and PtdIns(4,5)P$_2$ act by recruiting specific sets of pleckstrin homology domain-containing proteins to the plasma membrane (e.g. *Lietzke et al., 2000*), where they become active.

The best-studied functions of PTEN are as a negative regulator of proliferation and a positive regulator of apoptosis through the PDPK1-AKT-mTOR pathway (*Chalhoub and Baker, 2009*; *Song et al., 2012*). In addition to its role in tumorigenesis, loss of one copy of the wild-type *PTEN* gene leads to complex human developmental disorders such as Cowden and Bannayan-Riley-Ruvalcaba syndromes, which are characterized by macrocephaly, benign tumors, arteriovenous malformations, and autism spectrum disorder (*Blumenthal and Dennis, 2008*; *Zhou and Parada, 2012*). Phosphoinositides play important roles in the architecture of epithelia (*Shewan et al., 2011*), consistent with the high frequency of *PTEN* mutations in carcinomas. Studies on lumen morphogenesis in a three-dimensional culture system showed that PtdIns(4,5)P$_2$ is enriched in the apical membrane, whereas PtdIns(3,4,5)P$_3$ is enriched in basolateral membranes (*Martin-Belmonte et al., 2007*), and

**eLife digest** In mammals, the brain and spinal cord develop from a flat sheet of cells called the neural plate, which bends around to create a structure known as the neural tube. This bending process occurs through a complex sequence of cell shape changes. The cells in the neural plate are initially short and wide, but transform into long, thin cells as the neural plate forms. Problems that prevent the neural tube from forming correctly are amongst the most common birth defects in humans.

Many cancer cells contain a mutation that affects a gene that produces a protein called PTEN. This protein normally activates a tumor suppressor pathway, and so cancer cells that lack PTEN divide and grow uncontrollably. Grego-Bessa et al. have now examined mouse embryos that lack this gene, and found that the neural plate in such embryos forms irregular ruffles rather than a closed tube.

Further investigation revealed that the neural tube defects are not due to the inactivation of the traditional tumor suppressor pathway. Instead, correct neural tube formation relies upon the ability of PTEN to remove phosphate groups from a target lipid, which is important for limiting the activity of an enzyme called PDK1. Unlimited PDK1 activity causes complex changes that prevent the neural plate cells from elongating and packing together correctly. Future work is now needed to investigate the exact molecules targeted by PDK1 and the roles they play in disorders and diseases caused by a lack of the PTEN protein.

this was proposed to be important in tumor development (*Shewan et al., 2011*). Mammalian PTEN regulates cellular processes as diverse as collective cell migration (*Bloomekatz et al., 2012*) and axon regeneration (*Park et al., 2008*), and some of the effects of PTEN are independent of the AKT pathway (e.g. *Vasudevan et al., 2009*).

PTEN is essential for viability and *Pten* null mouse embryos arrest at midgestation with a complex set of morphological defects (*Suzuki et al., 1998*; *Bloomekatz et al., 2012*). We showed previously that PTEN is required for the directional collective migration of a population of extraembryonic cells, the anterior visceral endoderm (AVE), which must move from a distal to proximal position to define the anterior-posterior body axis of the embryo (*Bloomekatz et al., 2012*). PTEN is also required in the cells of the embryo proper: deletion of *Pten* in cells of the epiblast (the embryo proper) using the *Sox2-Cre* transgene (*Hayashi et al., 2002*) (*Pten* △Epi) bypasses the requirement for AVE migration but leads arrest at midgestation (~E9.0) with a syndrome of defects that included cardia bifida, abnormal mesoderm migration, and an abnormal open neural tube (*Bloomekatz et al., 2012*).

Mammalian neural tube closure requires more than 100 genes that regulate a sequence of orchestrated morphogenetic processes that transform the neural epithelium into a closed tube (*Copp and Greene, 2010*; *Harris and Juriloff, 2010*; *Colas and Schoenwolf, 2001*). Failure of any one of these events can cause neural tube defects, the second most common type of human birth defect after cardiac malformations. Most genetic studies of neural tube closure have focused on the cell rearrangements in the ventral midline mediated by the planar cell polarity pathway (*Murdoch et al., 2003*; *Ybot-Gonzalez et al., 2007*; *Nishimura et al., 2012*; *Williams et al., 2014*) or on the actin-mediated apical constriction of neural epithelial cells required for neural tube closure (*Suzuki et al., 2012*; *Grego-Bessa et al., 2015*). Prior to apical constriction, the neural plate lateral to the midline is transformed from a cuboidal to a tightly packed pseudostratified columnar epithelium, so that by E9.5, up to 8 nuclei are stacked on top of each other, with each cell retaining connections to both the apical surface and the basement membrane of the epithelium.

Here we define the cellular and biochemical basis of the neural tube closure defect seen in mouse embryos that lack PTEN. The *Pten* neural plate phenotype is not the result of changes in proliferation, apoptosis, cell fate or loss of epithelial polarity. Instead, *Pten* mutants have a novel defect in neural morphogenesis: they fail to form a pseudostratified columnar epithelium. Cells do not elongate along their apical-basal axis; they fail to become compacted along the mediolateral axis of the embryo and they fail to pack into a stable hexagonal array. A combination of genetic and chemical genetic experiments demonstrate that these defects are due to the loss of the lipid phosphatase activity of PTEN and to the activation of 3-phosphoinositide-dependent protein kinase-1 (PDPK1

(PDK1)), but do not depend on the AKT-mTOR tumor suppressor pathway. The data suggest that PTEN activity is required for stabilization of cell packing in the neural plate, which is in turn required for formation of apical-basal microtubule arrays, apical-to-basal trafficking, and cell elongation in the neural plate. We suggest that the role of PTEN in epithelial morphogenesis contributes to the developmental malformations in *PTEN* mutant syndromes and to the behavior of tumors that lack PTEN.

## Results

### PTEN is required for formation of the pseudostratified neural epithelium, but not for proliferation, patterning or apical-basal polarity

The cephalic neural epithelium in *Pten*[-/-] or *Pten* △Epi embryos does not close to make a neural tube (*Bloomekatz et al., 2012*). At E8.5, scanning electron micrographs showed that the wild-type cephalic neural plate was a smooth structure in which both sides have elevated to begin neural tube closure (*Figure 1A,C*). In contrast, irregular folds appeared in the *Pten* mutant neural plate as early as E8.0 and the neural plate was dramatically ruffled at E8.5 (*Figure 1B,D*); the position of the ectopic folds was highly variable between embryos. PTEN protein was strongly expressed in the E8.5 wild-type neural plate, where it was enriched both apically and basally (*Figure 1—figure supplement 1A–F*), consistent with a significant role for PTEN in morphogenesis of the neural tube. Phosphorylated AKT was not detectable in the wild-type neural plate, but was present in all membranes of *Pten* △Epi neural plate cells (*Figure 1—figure supplement 1G,H*), consistent with strong activation of the PI3 kinase pathway in *Pten* mutants.

The abnormal morphology of *Pten*[-/-] embryos was noted in previous experiments and was attributed to increased proliferation (*Stambolic et al., 1998*); however we previously showed that proliferation, cell number, and interkinetic nuclear migration are normal in the *Pten*[-/-] neural plate (*Bloomekatz et al., 2012*). Previous data suggested that there might be abnormalities in anterior-posterior patterning of cell types in the *Pten*[-/-] brain that could account for the abnormal morphology of the anterior neural tube (*Suzuki et al., 1998*). However, we found that anterior-posterior and dorsal-ventral neural patterning were normal in *Pten* △Epi embryos (*Figure 1—figure supplement 2A,B*). It has also been reported that loss of *Pten* activates canonical Wnt signaling (*Chen et al., 2015*), but expression of the canonical Wnt reporter TOPGAL was normal in *Pten* △Epi embryos (*Figure 1—figure supplement 2C*).

Transverse sections of the cephalic neural plate showed striking differences in organization in the wild-type and *Pten* △Epi cephalic neural epithelium. (For simplicity, we refer to *Pten* △Epi in the text below as *Pten*.) The wild-type neural plate is a single-layered columnar epithelium; the cells of the neural epithelium are so tightly packed that the nuclei appear to stack on top of each other, creating a pseudostratified epithelium. Nuclei in the cephalic neural plate, marked by expression of nuclear SOX2, were stacked in 3–5 rows at E8.5 (*Figure 1E*). In contrast, the SOX2+ nuclei of the E8.5 *Pten* cephalic neural plate were organized in only 1–3 rows (*Figure 1F*; *Figure 1—figure supplement 3I*).

Apical recruitment of PTEN is required for apical-basal polarity during apical lumen formation by MDCK cells (*Martin-Belmonte et al., 2007*). In contrast, we found that global apical-basal organization in the mouse neural plate was normal in the absence of PTEN. Laminin was basal, and F-actin, N-cadherin, ZO1, aPKC and Par3 were correctly localized to the apical domain in the mutant neural plate (*Figure 1G,H*; *Figure 1—figure supplement 3A–H*). Thus the data indicate that the *Pten* neural plate phenotype is not caused by abnormalities in proliferation, patterning or global apical-basal polarity; instead PTEN is required for normal morphogenesis of the neural plate.

### *Pten*[-/-] neuroepithelial cells are cuboidal rather than columnar and lack stable microtubule arrays

Because cells are very tightly packed in the neural plate, we used the mosaic expression of a cytoplasmic X-linked GFP transgene (*Hadjantonakis et al., 2001*) to visualize the shape of individual neural cells. In wild type, neural plate cells were highly elongated along the apical-basal axis, whereas *Pten* neuroepithelial cells were shorter and wider (*Figure 2A,B*). Accompanying the lack of pseudostratification, the *Pten* neural plate was 1.5 fold wider than the wild type: the mediolateral apical contour (from left to right) at the level of the mid-hindbrain junction in the E8.5 wild-type

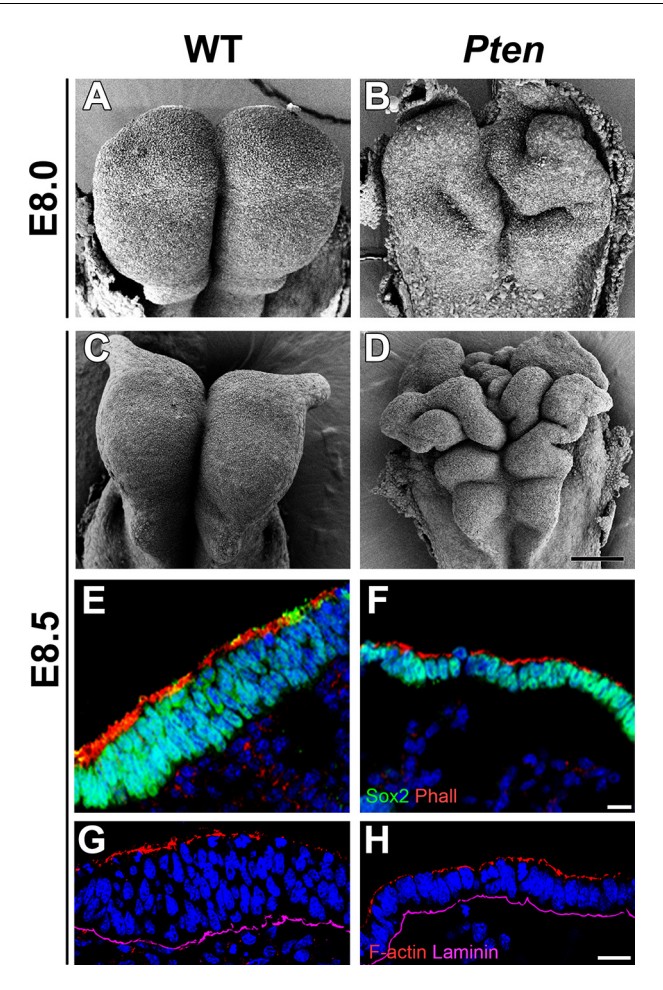

**Figure 1.** Morphological defects in the *Pten* mutant cephalic neural plate. (A, B, C, D) Comparison of neural plate morphology of the dorsal head of wild-type (WT) and *Pten* △Epi mutant embryos at E8.0 and E8.5 in scanning electron microscope images. Scale bar = 100 µm. (E, F) Transverse sections of E8.5 WT and *Pten* △Epi embryos show the absence of pseudostratified columnar organization in the *Pten* mutant cephalic neural plate. Green is SOX2, red is phalloidin (F-actin), blue is DAPI. (G, H) Z-stack projection of three optical sections (total of 3 µm) from transverse sections of the cephalic neural plate of E8.5 WT and *Pten* △Epi mutant embryos stained for phalloidin (red) and laminin (purple). Scale bar E–H = 10 µm.

The following figure supplements are available for figure 1:

**Figure supplement 1.** PTEN expression in the cephalic neural plate.

**Figure supplement 2.** Normal neural patterning in *Pten△Epi* embryos.

**Figure supplement 3.** Apical markers in *Pten* △Epi mutant embryos.

neural plate was $668 \pm 242$ µm wide (n = 6) and $1023 \pm 369$ µm wide in *Pten* (n = 6). Despite this increase in width, the number of nuclei across the width of the cephalic neural plate was not changed in the mutant ($275 \pm 140$ nuclei wide in wild type; $262 \pm 88$ nuclei in *Pten* mutants), indicating that the same number of cells occupy more area in *Pten*.

We measured the apical surface area of individual neural plate cells by *en face* imaging, with cell boundaries marked by expression of the tight junction marker ZO1 (*Figure 2C*). At the onset of neural morphogenesis (head fold stage, E7.75), the apical surfaces of wild-type and *Pten* mutant cells were both variable in size and shape but had the same average area (approximately 30 µm²;

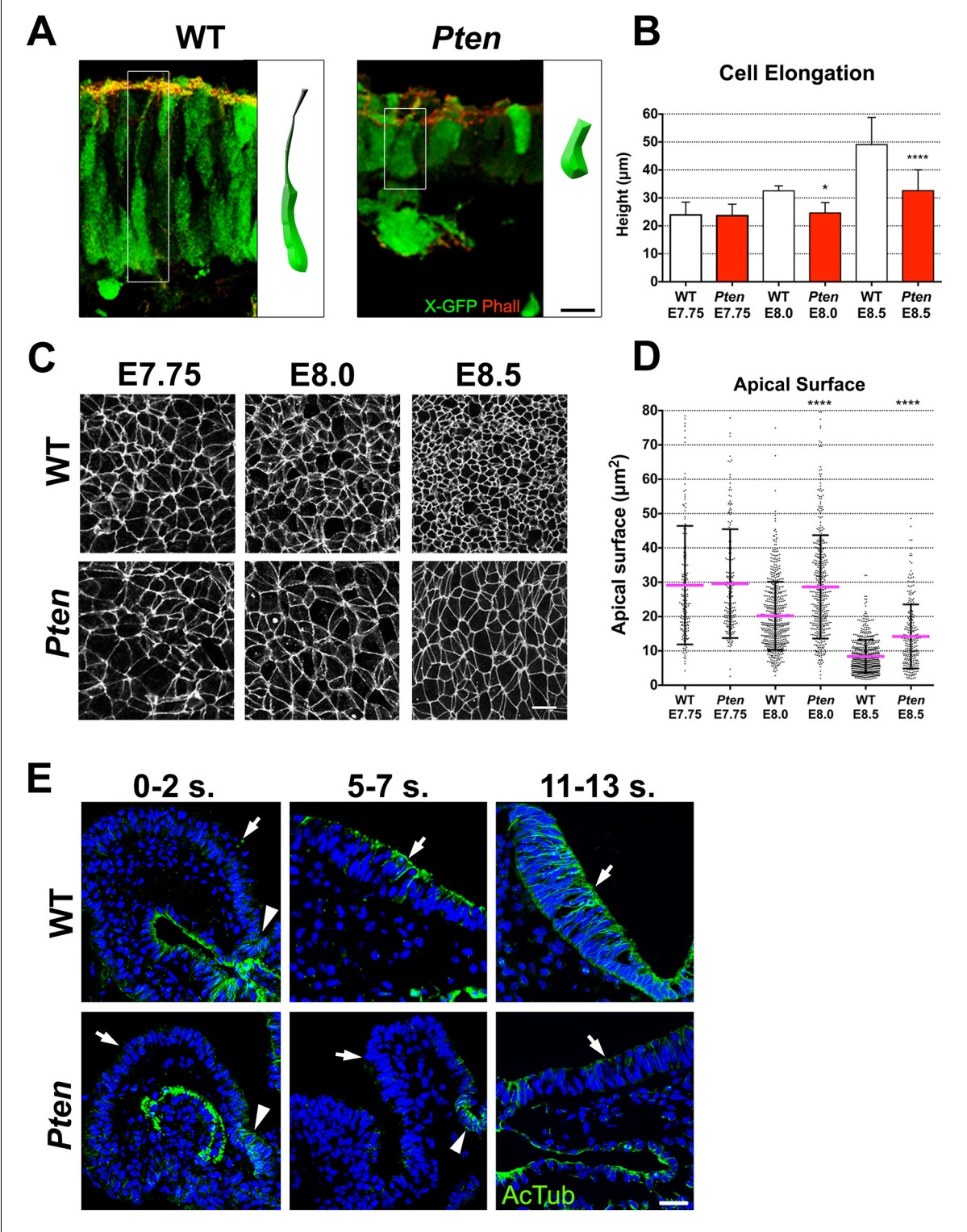

**Figure 2.** Cellular defects of *Pten* △Epi mutant neuroepithelial cells. (**A**) Comparison of WT and mutant cell shape in the E8.5 cephalic neural plate, using X-linked GFP-expression to mark individual cells. Schematic representations of individual cells for each genotype are shown (white box). Red is phalloidin. Scale bar is 10 µm. (**B**) Comparison of neural plate height in the cephalic region of WT and mutants. WT E7.75 = 23.9 ± 4.5 µm; *Pten* △Epi E7.75 = 23.6 ± 4.1 µm: WT and mutant are not different, p = 0.86, by standard t-test. WT E8.0 = 32.5 ± 1.7 µm; *Pten* △Epi E8.0 = 24.6 ± 3.7 µm: WT is significantly taller than the mutant, *p = 0.0164. WT E8.5 = 49.1 ± 9.6 µm; *Pten* △Epi E8.5 = 32.6 ± 7.4 µm; WT is significantly taller than the mutant, ****p < 0.0001. For this and similar analyses below, >100 measurements were made from ≥3 embryos. (**C**) Comparison of apical cell shape in the

*Figure 2 continued on next page*

*Figure 2 continued*

cephalic neural epithelium of WT and *Pten* △Epi embryos viewed *en face* at E7.75, E8.0 and E8.5. Cell borders are marked by expression of ZO1 (white). Scale bar = 20 μm. (D) Apical surface of cephalic neural epithelial cells, taken from images like those shown in (C). WT E7.75 = 29 ± 17 μm$^2$; *Pten* △Epi E7.75 = 30 ± 16 μm$^2$: WT and mutant are not different, p = 0.79. WT E8.0 = 20 ± 10 μm$^2$; *Pten* △Epi E8.0 = 29 ± 15 μm$^2$. The WT surface area is significantly smaller than in the mutant, ****p < 0.0001. WT E8.5 = 8 ± 4 μm$^2$; *Pten* △Epi E8.5 = 14 ± 9 μm$^2$. The WT surface area is significantly smaller than in the mutant, ****p < 0.0001. (E) Acetylated microtubule arrays in the neural plate in stage-matched WT and mutant embryos. Transverse sections of cephalic regions of WT and *Pten* △Epi embryos at E8.0 (0– 2 somites), E8.5 (5–7 somites) and E9.0 (11–13 somites). Green is acetylated tubulin; blue is DAPI. Arrows point to the apical surface of neural plate; arrowheads point to the floor plate. The first region of tubulin acetylation in WT is in the floor plate, which is only region of tubulin acetylation in the mutant. Scale bar = 25 μm.

The following figure supplement is available for figure 2:

**Figure supplement 1.** Acetylated microtubules in the wild type cranial neural plate.

*Figure 2D*). By ~6 hr later, at E8.0, the average apical surface area of wild-type neural cells had decreased to ~20 μm$^2$, whereas the apical surface area of *Pten* mutant cells was unchanged (*Figure 2D*). At E8.5, the apical surface of wild-type neural plate cells had shrunk further, so that it was ~8 μm$^2$, ~3.5 fold smaller than at E7.75. Between E8.0 and E8.5, the surface area of *Pten* neural plate cells also decreased, but the area of mutant cells was still ~40% greater than that of wild type (*Figure 2D*). At the same time as the apical surface of wild-type neural plate cells decreased, cell volume remained constant, so the height of the cells increased ~2 fold in WT embryos from ~24 μm at E7.75 to ~50.0 μm at E8.5 (*Figure 2B*), while the height of *Pten* mutant cells increased only ~1.3 fold, to ~30 μm at E8.5 (*Figure 2B*).

Formation of polarized columnar epithelia is accompanied by the formation of arrays of apicobasally polarized stable microtubules, with minus-ends apical (*Bré et al., 1987*; *Jaulin and Kreitzer, 2010*). For example, in the neural plate of the *Xenopus* embryo, multiple γ-tubulin-positive apical centrioles nucleate stable arrays of parallel, acetylated microtubules that are thought to drive elongation of the cells along the apical-basal axis (*Lee et al., 2007*). In cells of the mouse embryo neural plate, there is only a single apical centrosome, but noncentrosomal microtubule arrays, marked by expression of α-tubulin, were present parallel to the apical-basal axis of cephalic neural plate cells in both the E8.5 wild-type neural plate, although α-tubulin arrays were not apparent in the *Pten* mutant (*Figure 2—figure supplement 1A*). Stable microtubules can become acetylated (*Palazzo et al., 2003*); wild-type microtubule arrays were not acetylated at E8.0 (0–2 somites) except in the floor plate but became acetylated by E8.5 (5–7 somites) and were strongly acetylated at E9.0 (11–13 somites) (*Figure 2E*; *Figure 2—figure supplement 1B,C*). In contrast, the *Pten* neural plate lacked acetylated microtubules at each of these stages; the only acetylated microtubule arrays in the mutant neural plate were located in the floor plate, the cells in the ventral midline (*Figure 2E*).

## Constitutive activation of PI3 kinase recapitulates the *Pten* neural plate phenotype

Because PTEN has both lipid and protein phosphatase activities (*Worby and Dixon, 2014*), we tested whether the lipid phosphatase activity of PTEN mediated its role in epithelial morphogenesis. While PTEN dephosphorylates PtdIns(3,4,5)P$_3$ to PtdIns(4,5)P$_2$, phosphoinositide 3-kinase (PI3 kinase) carries out the reverse reaction and produces PtdIns (3,4,5)P$_3$. We injected the pregnant mothers of *Pten* mutant embryos at E7.5 with LY294002, a small molecule inhibitor of PI3 kinase (*Gharbi et al., 2007*) and analyzed the embryonic phenotype 24 hr later. The development of wild-type embryos was not affected by this treatment, but the mutant neural plate appeared rescued: it was pseudostratified and showed acetylated microtubules arrays (*Figure 3—figure supplement 1*). Thus inhibition of PI3 kinase rescued *Pten* neural plate phenotype, suggesting that it is the lack of the lipid phosphatase activity that causes the *Pten* mutant phenotype.

We used an independent genetic experiment to test whether increased levels of PtdIns(3,4,5)P$_3$ were responsible for the defects in epithelial morphogenesis. *Pik3ca* encodes the p110 catalytic subunit of PI3 kinase that catalyzes the production of PtdIns(3,4,5)P$_3$. Point mutations in *PIK3CA* are seen frequently in tumors and approximately 40% of breast cancer *PIK3CA* mutations are due to a single amino acid substitution allele, *PIK3CA*$^{H1047R}$, which causes elevated kinase activity (*Saal, 2005*;

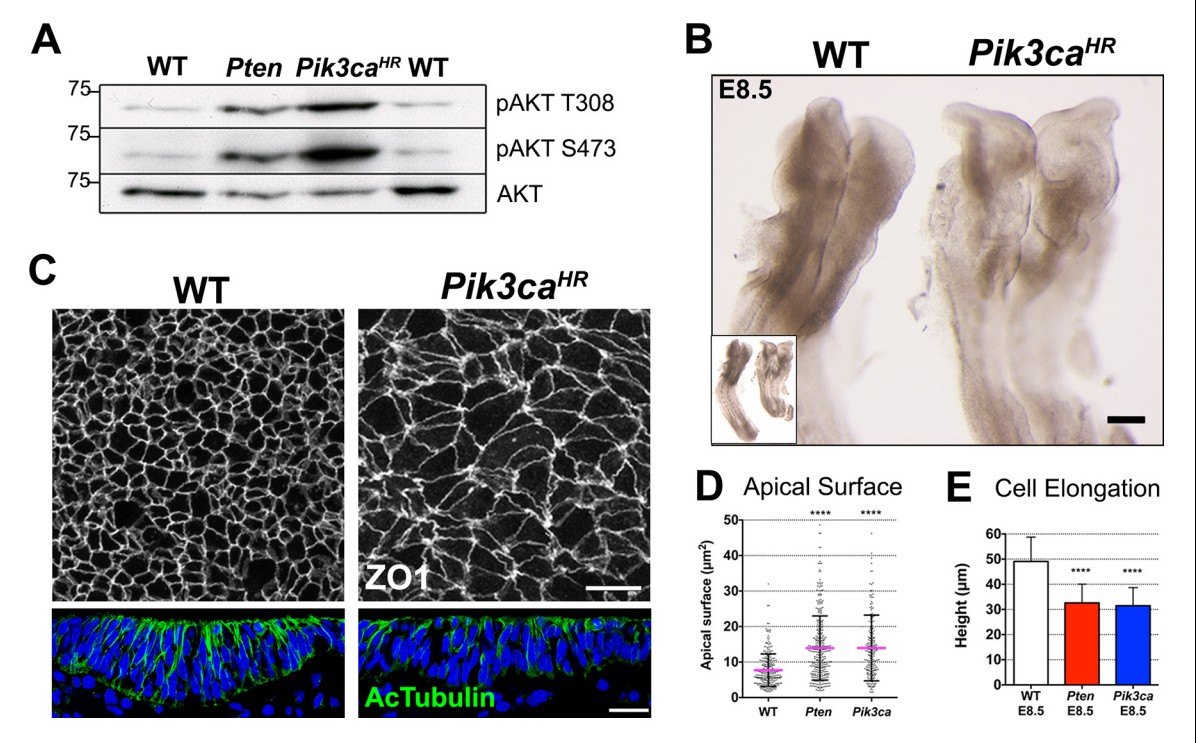

**Figure 3.** Expression of an activated form of PI3 Kinase mimics the *Pten* mutant neural plate phenotype. (**A**) Loss of *Pten* (*Pten* △Epi) or expression of the activating mutation *Pik3ca*<sup>H1047R</sup>-Epi in the epiblast leads to phosphorylation of AKT in E8.5 embryos. Representative Western blots (n = 3) show the two phosphorylated forms of AKT in WT, *Pten* △Epi and *Pik3ca*<sup>H1047R</sup>–Epi embryos. Numbers indicate approximate MW. (**B**) *Pik3ca*<sup>H1047R</sup>–Epi embryos phenocopy *Pten* △Epi embryos. Whole embryos (inset) and expanded view of the cephalic region of E8.5 WT and *Pik3ca*<sup>H1047R</sup>-Epi embryos; dorsal view. Scale bar = 120 μm. (**C**) The apical surface of the neural plate, viewed *en face*; cell borders marked by expression of ZO1 (white) (top row), and acetylated tubulin (green) in transverse sections of the cephalic neural epithelium of E8.5 WT and *Pik3ca*<sup>H1047R</sup>-Epi embryos. Blue is DAPI. Scale bar = 20 μm. (**D**) Comparison of apical surface area of cephalic neural epithelial cells at E8.5. WT = 8 ± 4 μm$^2$; *Pten* △Epi = 14 ± 9 μm$^2$; *Pik3ca*<sup>H1047R</sup>-Epi = 15 ± 10 μm$^2$. The surface areas of both mutants are significantly larger than wild type, ****p < 0.0001. (**E**) Cephalic neural plate height at E8.5. WT = 49.1 ± 9.6 μm; *Pten* △Epi = 32.6 ± 7.4 μm; *Pik3ca*<sup>H1047R</sup>-Epi = 31.5 ± 7.2 μm. Cells in both mutants are significantly shorter than in wild type, ****p < 0.0001.

The following figure supplement is available for figure 3:

**Figure supplement 1.** Inhibition of PI3 kinase restores pseudostratification in the *Pten* △Epi neural plate.

*Carson et al., 2008*). We conditionally expressed a *Pik3ca*<sup>H1047R</sup> allele in the epiblast under the control of the *Sox2* promoter (*Pik3ca*<sup>H1047R</sup>-Epi). Western blot analysis confirmed that both pAKT Thr308 and pAKT Ser473, well-characterized targets of the PI3-kinase pathway (*Sarbassov et al., 2005*), were elevated in both *Pten* and *Pik3ca*<sup>H1047R</sup>-Epi embryos (*Figure 3A*).

*Pik3ca*<sup>H1047R</sup>-Epi embryos had an open, ruffled cephalic neural plate, similar to that seen in *Pten* △Epi (*Figure 3B*). Transverse sections of the cephalic neural plate showed that *Pik3ca*<sup>H1047R</sup>-Epi neural plate cells did not become columnar (height of E8.5 neural plate cells = 31.5 ± 7.2 μm), the nuclei failed to become pseudostratified, and there was reduced expression of acetylated tubulin (*Figure 3C*). The apical surface area of E8.5 *Pik3ca*<sup>H1047R</sup>-Epi neural plate cells was ~15 μm$^2$, ~40% larger than wild type (*Figure 3C,D*), and epithelial cell height was ~40% shorter than in wild type, as seen in *Pten* (*Figure 3C,E*). The common defects in *Pik3ca*<sup>H1047R</sup>-Epi and *Pten* △Epi embryos argue that elevated levels of PtdIns(3,4,5)P$_3$ were responsible for the neural plate phenotypes of both mutants.

## Removal of 3-phosphoinositide dependent protein kinase 1 (*PDPK1*) rescues the *Pten* neural plate phenotype

In the PTEN tumorigenesis pathway, elevated PtdIns(3,4,5)P$_3$ recruits 3-phosphoinositide-dependent protein kinase-1 (PDPK1) to the plasma membrane through its PH domain, thereby allowing PDPK1 access to specific substrates, including AKT, an important target in tumorigenesis (*Sommer et al., 2013*). *Pdpk1* null embryos die at midgestation with defects in morphogenesis of the brain and somites; proliferation and apoptosis are normal in null mutant MEFs, but *Pdpk1* mutant cells are small (*Lawlor et al., 2002*).

To assess the role of *Pdpk1* in neural morphogenesis, we removed the gene in embryonic line-ages using a conditional *Pdpk1* allele with *Sox2-Cre (Pdpk1* △Epi). The general morphology of *Pdpk1* △Epi embryos was similar to that previously described for the *Pdpk1* null allele (*Lawlor et al., 2002*), although the conditionally deleted embryos appeared more healthy, formed recognizable somites and initiated embryonic turning, unlike the null mutants. The sides of the neural plate in *Pdpk1* △Epi failed to elevate at E8.5, but the neural tube closed by E9.5 (*Figure 4—figure supplement 1A,B*). Transverse sections at E8.5 and E9.5 showed multiple layers of nuclei and strong acetylated tubulin staining in cephalic neural tube (*Figure 4—figure supplement 1A,B*), indicating that cell elongation and neural plate pseudostratification occurred in absence of PDPK1.

To test whether the neural morphogenesis defects observed in *Pten* neural plate required the activity of PDPK1, we simultaneously removed both *Pdpk1* and *Pten* in the epiblast using the *Sox2-Cre* transgene. While pAKT levels were increased in *Pten* embryos, the levels of both phosphory-lated forms of AKT were decreased in *Pdpk1* △Epi single mutants (hereafter referred to as *Pdpk1*) and were present at approximately normal levels in *Pten* △Epi *Pdpk 1*△Epi double mutant embryos (referred to below as *Pten Pdpk1* double mutants) (*Figure 4A*). Phosphorylation of the AKT target GSK3β (Ser9) was decreased (*Figure 4A*), confirming that activation of AKT by removal of PTEN depends on PDPK1, as in other cell types. We noted that phosphorylation of the downstream target ribosomal protein S6 was not affected in *Pten* embryos, while phosphorylation of S6 was abolished in *Pdpk1* single and *Pten Pdpk1* double mutant embryos (*Figure 4A*). The absence of increased phosphorylation of S6 in *Pten* embryos probably reflects the high rates of growth and cell division in the wild-type mouse embryo, which are not further increased by removal of PTEN.

The global morphology of the *Pten Pdpk1* double mutant embryos resembled that of the *Pdpk*1 single mutants (*Figure 4B*). The cells in the E8.5 double mutant cephalic neural plate were elongated similar to wild type (E8.5 *Pten Pdpk1* neural plate height = 48.6 ± 8.8 μm), pseudostratified, and there were apical-basal arrays of acetylated microtubules in the double mutant neural plate (*Figure 4C,D*). The apical surface area of cells in E8.5 *Pten Pdpk1* double mutant neural plate was 50% less than in *Pten* embryos (10 ± 7 μm$^2$ compared to 15 ± 9 μm$^2$), indicating a rescue of cell shape (*Figure 4C,E*). Thus these aspects of the *Pten* neural plate phenotype depend on PDPK1.

PTEN acts in extraembryonic tissues to control polarized collective migration of the anterior vis-ceral endoderm that establishes the anterior-posterior body axis and in the epiblast to control move-ment of cardiac precursor cells to the midline (*Bloomekatz et al., 2012*). In double mutants that lack both *Pten* and *Pdpk1* in all tissues (*Pten$^{-/-}$; Pdpk1$^{-/-}$*), the embryos showed the partial axis duplication seen in *Pten* single mutants (*Figure 4—figure supplement 2A*). *Pten* △Epi *Pdpk1* △Epi double mutants showed the cardia bifida phenotype seen in *Pten* △Epi embryos (*Figure 4—figure supple-ment 2B*). Thus these cell migration phenotypes in *Pten* mutant embryos were not rescued by removal of PDPK1, in contrast to the PDPK1-dependent phenotype of the *Pten* neural plate.

## The neural plate defects in Pten mutants are independent of AKT and mTORC1

AKT is a direct substrate for phosphorylation by PDPK1 (*Walker et al., 1998*) and the biochemical assays showed that AKT phosphorylation was increased in *Pten* mutant embryos (e.g. *Figure 3A*), as expected. There are three *Akt* genes in the mouse with overlapping functions (*Gonzalez and McGraw, 2009*), prohibiting a classical genetic test of the role of *Akt* in neural morphogenesis. Therefore to test whether pAKT was required for the *Pten* △Epi phenotype, we injected mothers of *Pten* mutant embryos with MK-2206, an allosteric inhibitor that blocks activation of the three AKT isoforms (*Hirai et al., 2010*), 24 and 48 hr before embryo dissection. Western blot analysis showed

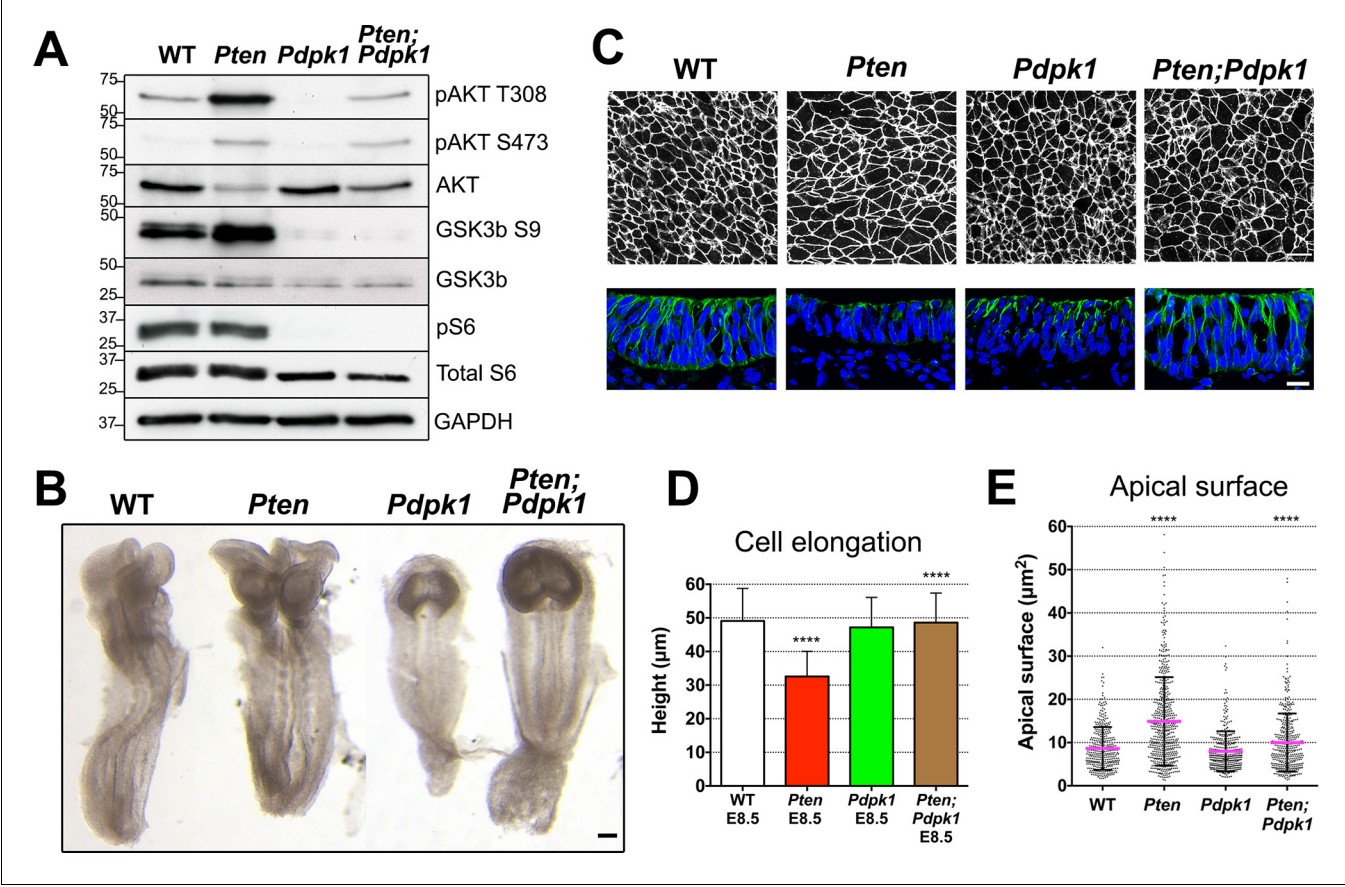

**Figure 4.** Removal of *Pdpk1* rescues the pseudostratified columnar organization of the *Pten* neural plate. (**A**) Phosphorylation of downstream targets of the PI3 kinase pathway in E8.5 wild type, *Pten* △Epi, *Pdpk1* △Epi single mutant and *Pten* △Epi *Pdpk1* △Epi double mutant embryos. Representative western blot shown (n = 3). Numbers indicate approximate MW. (**B**) Dorsal views of E8.5 wild-type, *Pten* △Epi, *Pdpk1*△Epi and *Pten* △Epi *Pdpk1* △Epi embryos. The *Pten Pdpk1* double mutants are similar in morphology to *Pdpk1* single mutants, but are larger. Scale bar = 100 μm. (**C**) The apical surface of the neural plate, viewed *en face*. Cell borders marked by expression of ZO1 (white) (top row) and acetylated tubulin (green) in transverse sections of cephalic neural epithelium in E8.5 wild-type, *Pten* △Epi, *Pdpk1* △Epi and *Pten* △Epi *Pdpk1* △Epi embryos. Blue is DAPI. Scale bar = 20 μm. (**D**) Cephalic neural plate height at E8.5. WT = 49.1 ± 9.6 μm; *Pten* △Epi = 32.6 ± 7.4 μm; *Pdpk1* △Epi = 47.2 ± 8.9 μm; *Pten* △Epi *Pdpk1* △Epi = 48.6 ± 8.8 μm. *Pten* △Epi cells are significantly shorter than in wild type, and *Pten* △Epi *Pdpk1* △Epi double mutant cells are significantly taller than in *Pten* △Epi, ****$p < 0.0001$. (**E**) Apical surface area of E8.5 cephalic neuroepithelial cells. Wild type = 9 ± 6 μm²; *Pten* △Epi = 15 ± 9 μm². The surface area of *Pten* △Epi is significantly greater than in wild type, ****$p < 0.0001$; *Pdpk1* △Epi = 8 ± 5 μm²; *Pten* △Epi *Pdpk1* △Epi = 10 ± 7 μm²; the surface area of *Pten* △Epi *Pdpk1* △Epi double mutant cells is significantly less than in *Pten* △Epi, ****$p < 0.0001$.

The following figure supplements are available for figure 4:

**Figure supplement 1.** The *Pdpk1*△Epi phenotype.

**Figure supplement 2.** Cell migration phenotypes in *Pten Pdpk1* double mutants.

that the treatment effectively blocked phosphorylation of AKT on both Thr308 and Ser473 (*Figure 5A*).

Despite effective inhibition of AKT activation, treatment with MK-2206 had no detectable effect on the morphology of the neural plate of E8.5 *Pten* embryos (*Figure 5B*). *En face* imaging and transverse sections showed that blocking AKT activity did not rescue the neural plate height, pseudostratification or microtubule acetylation (*Figure 5C,D*). Quantification of apical surface area showed no significant difference between treated and untreated *Pten* embryos (*Figure 5E*).

An important downstream target of AKT is mTORC1, which mediates its effects on growth and survival (*Zoncu et al., 2011*). To test whether mTORC1 activity plays a role in morphogenesis of the

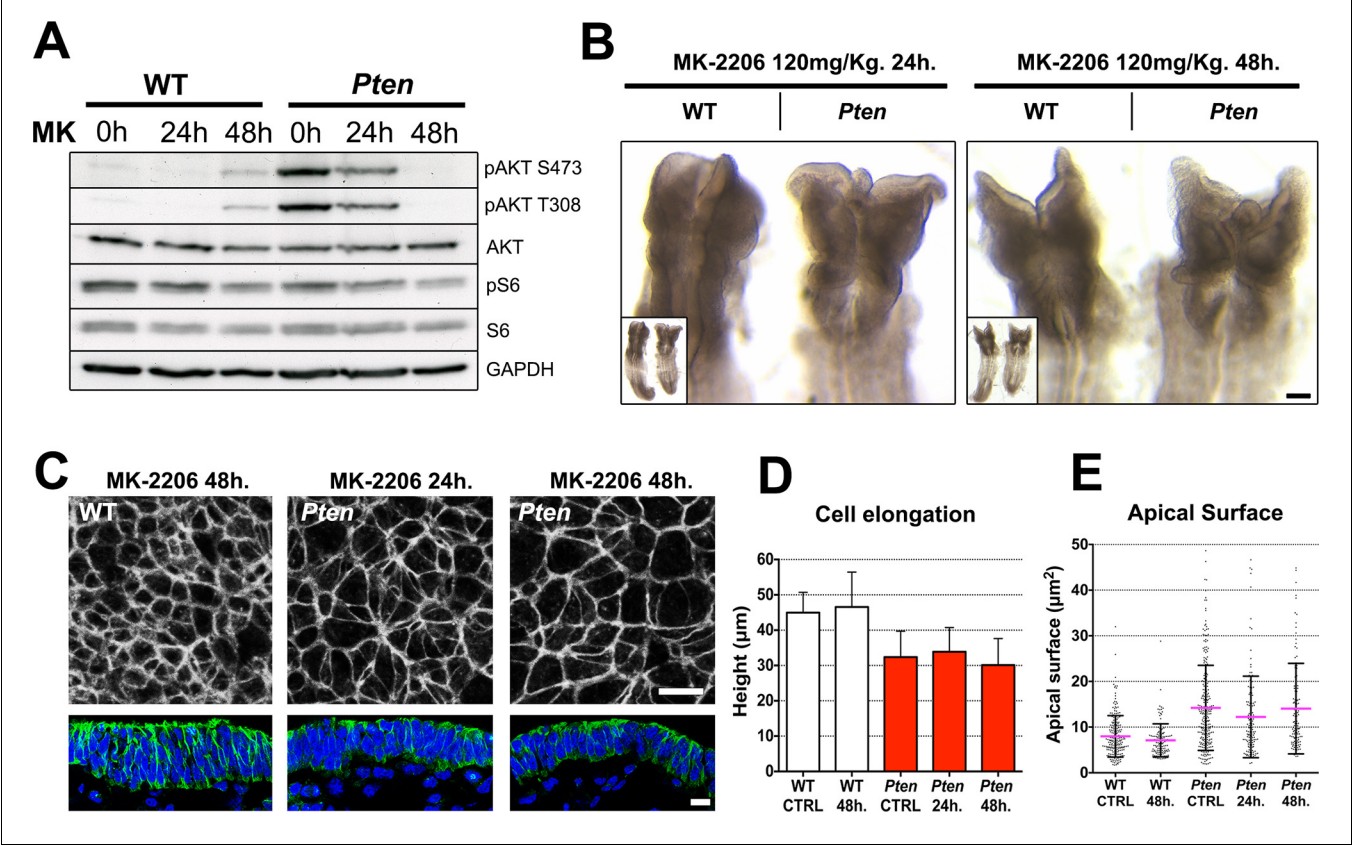

**Figure 5.** The *Pten* neural plate phenotype is independent of AKT. (**A**) Effect of the AKT inhibitor MK-2206 treatment on targets of the PI3 kinase pathway in E8.5 embryos. Western blot of the two phosphorylated forms of AKT and pS6 S240/4 in WT and *Pten* △Epi at E8.5 in control embryos (vehicle) and after 24 or 48 hr of MK-2206 treatment in utero prior to embryo dissection. Numbers indicate approximate MW. (**B**) Dorsal view (inset) and enlarged image of the cephalic region of E8.5 wild-type and *Pten* △Epi embryos. There is no change in the morphology of the mutant heads after 24 or 48 hr of MK-2206 treatment in utero. Scale bar = 120 μm. (**C**) The apical surface of the neural plate, viewed *en face*. Cell borders marked by expression of ZO1 (white) (top row); acetylated tubulin (green) in transverse sections of cephalic neural epithelium in wild type and *Pten* △Epi at E8.5 after 24 or 48 hr of MK-2206 treatment in utero. Blue is DAPI. Scale bar = 10 μm. (**D**) Height of the E8.5 cephalic neural plate. Wild type, untreated (control) = 44.9 ± 5.7 μm; WT 48 hr treatment = 46.5 ± 9.9 μm; MK-2206 treatment had no significant effect. *Pten* △Epi untreated (control) = 32.4 ± 7.3 μm; *Pten* △Epi 24 hr = 33.9 ± 6.8 μm²; *Pten* △Epi 48 hr = 30.3 ± 7.5 μm. Treated and untreated mutants were all significantly shorter than wild type, but MK-2206 treatment did not significantly rescue cell elongation in the mutant. (**E**) Apical surface area of E8.5 cephalic neuroepithelial cells. Control = 8 ± 5 μm²; WT 48 hr = 7 ± 4 μm²; *Pten* △Epi Control = 14 ± 9 μm²; *Pten* △Epi 24 hr = 13 ± 9 μm²; *Pten* △Epi 48 hr = 14 ± 10 μm². Treated and untreated mutant cells all had significantly larger surface area than wild type, but MK-2206 treatment did not significantly decrease cell surface area in the mutant.

The following figure supplements are available for figure 5:

**Figure supplement 1.** Inhibition of mTORC1 by rapamycin does not rescue the *Pten* neural plate phenotype.

**Figure supplement 2.** Downstream targets of PDPK1.

**Figure supplement 3.** Myosin-II distribution and levels appear normal in the Pten neural plate.

neural plate, we injected pregnant females with the mTor inhibitor rapamycin. Western blot analysis of treated embryos showed that the rapamycin treatment blocked phosphorylation of ribosomal protein S6, as expected (*Figure 5—figure supplement 1A*). Despite its clear biochemical activity, rapamycin did not rescue the cell shape, pseudostratification or tubulin acetylation in the *Pten* △Epi neural plate (*Figure 5—figure supplement 1B*). Thus neither AKT nor mTORC1 mediated the effect of PDPK1 on neural morphogenesis.

Many other direct substrates for phosphorylation by PDPK1 are known, including more than 20 protein kinases of the AGC family, in addition to AKT (*Pearce et al., 2010*). Atypical PKC (aPKC) and

PKN family members are PDPK1 targets that are stimulated through PtdIns(3,4,5)P$_3$ association (*Balendran et al., 2000*), and aPKC is an important regulator of epithelial polarity. However, we did not detect a change in localization or increased phosphorylation of aPKC in *Pten* mutants (*Figure 1—figure supplement 3*; *Figure 5—figure supplement 2A*). The Serum and Glucocorticoid-induced Kinase (SGK) protein family is also activated by phosphorylation by PDPK1. Phosphorylation of NDRG1 (T346) is mediated by SGK activity (*Murray et al., 2004*), and pNDRG1 was upregulated in *Pten* embryos and reduced in *Pten Pdpk1* double mutants (*Figure 5—figure supplement 2A*). However, in utero treatment of *Pten* embryos with the AKT inhibitor MK-2206 blocked phosphorylation of NDRG1 (*Figure 5—figure supplement 2B*), suggesting that activation of NDRG1 depends AKT and not on the pathway that regulates neural morphogenesis. Evidence suggests that PDPK1 can activate Rho kinase 1 (ROCK1) and phosphorylation of myosin light chain (*Pinner and Sahai, 2008*), which should increase the formation of myosin cables. However, myosin-II was anisotropically distributed in neural plate cells of all genotypes (wild type, *Pten$^{-/-}$*, *Pdpk1$^{-/-}$* and *Pten$^{-/-}$ Pdpk1$^{-/-}$*), there was no preferential enrichment of myosin-II at long or short cell edges in E8.0 embryos (*Figure 5—figure supplement 3A–D*) and phosphorylation of myosin light chain (MLC) was similar in wild type and *Pten* mutants (*Figure 5—figure supplement 3E*).

## PTEN and PDPK1 regulate cell packing in the neural plate

To define the cellular processes regulated by PDPK1 in the neural plate, we examined the cellular basis of the *Pten* mutant phenotype at higher resolution. Pten has been implicated in planar topology of epithelial cells in *Drosophila* (*Bardet et al., 2013*) and cells in the amniote neural plate undergo dynamic cellular reorganization during neural tube closure as cells break and remake junctions with their neighbors (*Schoenwolf and Alvarez, 1989*; *Nishimura et al., 2012*). In stable epithelia, cells are hexagonally packed into a honeycomb-like array: each cell has six neighbors and three cells converge on each vertex (*Zallen and Zallen, 2004*). In dynamic epithelia, this pattern can be disrupted by cell division or by neighbor exchanges, so that each cell has fewer neighbors and a greater number of cells converge on each vertex (*Zallen and Zallen, 2004*).

Visualizing cell borders with ZO1 (*Figure 2C*), β-Catenin (*Figure 6A*) or F-actin (*Figure 6D*; *Figure 6—figure supplement 1A,D*), cells at the beginning of wild-type neural morphogenesis (E8.0) were not hexagonally packed: only ~45% had five or six edges (*Figure 6B*). Cell arrangements included rosette-like structures where as many as 8 cells converged at a single vertex (*Figure 6A*), similar to structures in epithelia undergoing active cell rearrangements (*Blankenship et al., 2006*) and previously described in the rearranging cells of the neural floor plate in chick and mouse embryos (*Nishimura et al., 2012*; *Williams et al., 2014*). The arrangement of cells in the *Pten* neural plate at E8.0 showed the same organization as seen in wild type, where ≥4 cells converging on ~60% of the vertices (*Figure 6C*). At E8.5, when pseudostratification was apparent, cells in the wild-type neural plate were packed in a more honeycomb-like arrangement: ~1.8 fold more cells with 5 and 6 edges, and the percentage of cases with ≥4 cells converging on a vertex was reduced by half (to ~30%), consistent with a more stable epithelium (*Figure 6A–C*). In contrast, these parameters did not change between E8.0 and E8.5 in *Pten* mutants. Thus PTEN appears to promote a more regular, hexagonal organization in the plane of the epithelium at the same stage when the epithelium becomes columnar. Cells in the E8.5 neural plate cells of the constitutively activate PI3 kinase mutant (*Pik3ca$^{H1047R}$*-Epi) showed the complex cells arrangements and rosettes seen in *Pten* mutants (*Figure 6—figure supplement 1A–C*).

The organization of E8.5 *Pdpk1* single and the *Pten Pdpk1* double mutant neural plates were similar to wild type, with similar distributions of neighbors per cell (~60% of cells with 5 or 6 edges) and the percentage of cases with ≥4 cells converging on a vertex was ~30% (*Figure 6D–F*). Blocking AKT activity with MK-2206 did not modify cell packing in the *Pten* neural plate (*Figure 6—figure supplement 1D–F*). Thus, as with cell elongation and pseudostratification, the failure of *Pten* mutant neural plate to assume a stable conformation was caused by elevated PtdIns(3,4,5)P$_3$, and depended on PDPK1 but not AKT.

## PTEN and PDPK1 regulate apical-to-basal trafficking in the neural plate

The bottle cells of the gastrulating *Xenopus* embryo share some characteristics with the early neural plate: they begin as cuboidal cells that elongate in an apical-basal direction while forming apical-

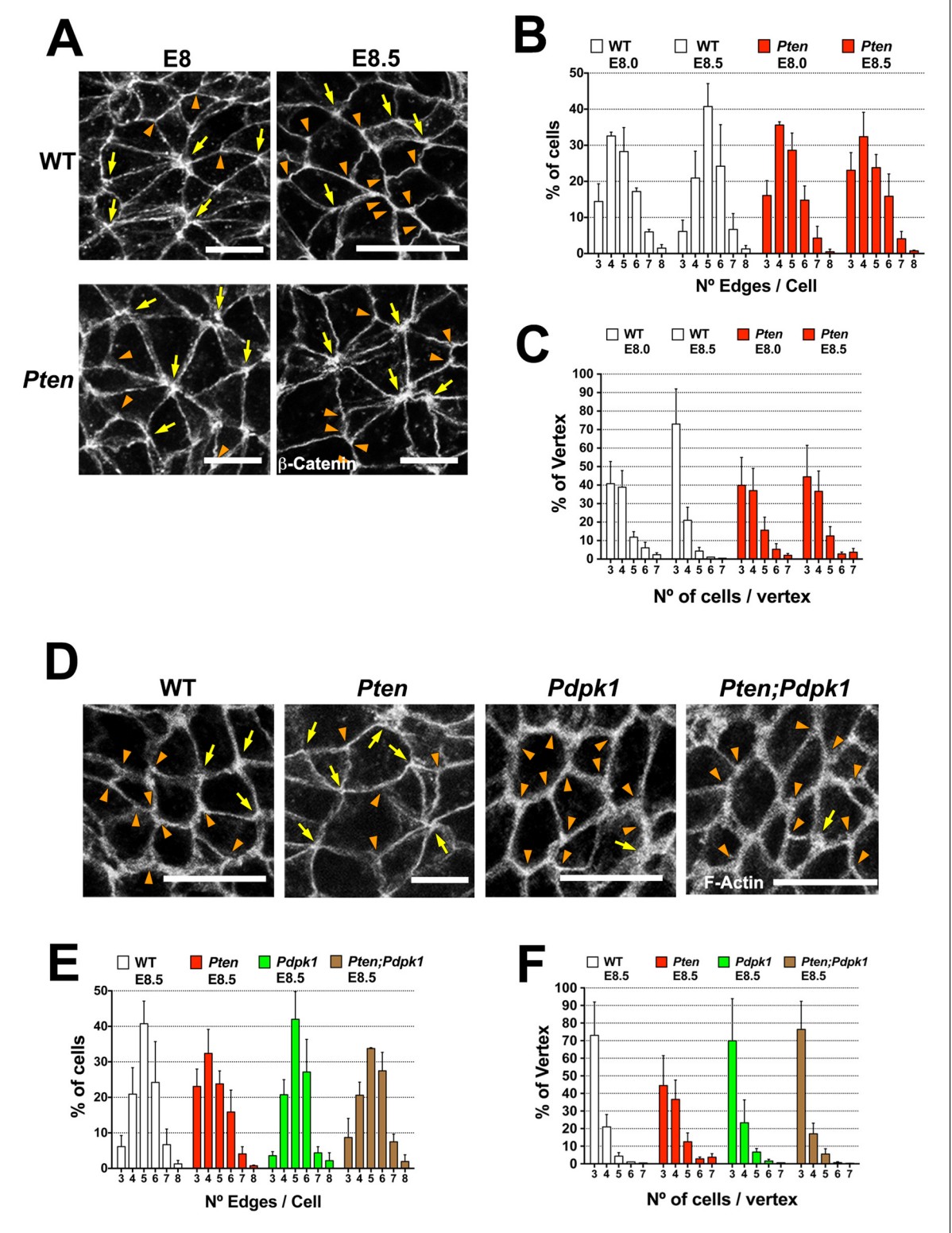

**Figure 6.** PTEN promotes stable cell packing in the neural plate. Panels (**A**) and (**D**) show high magnification views of the apical surface of the neural plate embryos, with magnification adjusted so that the cells appear to be approximately the same size, in order to highlight the difference in cell packing in the two genotypes. Scale bars in (**A**) and (**D**) = 15 μm. Orange arrowheads indicate examples of 3 cells/vertex, and yellow arrows indicate vertices formed by ≥4 cells. Cell borders marked by β-catenin (**A**) or F-actin (**D**) expression. (**A**) At E8.0, rosette-like structures are common in both WT and *Pten*. Fewer rosette-like arrangements are seen in WT at E8.5, but rosettes persist in the E8.5 *Pten* neural plate. (**B**) Quantification of percentage of

*Figure 6 continued on next page*

*Figure 6 continued*

cells with 3–8 edges. Between E8.0 and E8.5, the percentage of cells with 3–4 edges decreases ~45%, while the percentage with 5–6 edges increases ~1.6 fold in WT embryos, but these parameters are unchanged in E8.5 mutants. (**C**) The percentage of vertices plotted against the number of cells meeting at a vertex. In a honeycomb arrangement, 3 cells meet at a vertex; the number of cases where three cells meet at a vertex increases ~1.8 fold between E8.0 and E8.5, whereas the *Pten* neural plate does not changed in this interval. (**D**) At E8.5, *Pdpk1* single and *Pten Pdpk1* double mutants show packing similar to that in WT, compared to the more rosette-like packing in *Pten*. Quantification of % of cells with 3–8 edges (**E**) and % of vertices formed by 3–7 cells (**F**) showed similar values in E8.5 WT, *Pdpk1* and *Pten Pdpk1* embryos. Bars indicate %, lines indicate s.d.

The following figure supplement is available for figure 6:

**Figure supplement 1.** Cell packing in the neural plate with constitutively active PI3 kinase and when AKT is inhibited with MK-2206.

basal arrays of microtubules and constricting their apical surfaces (*Keller et al., 2003*; *Lee and Harland, 2007*). During the cuboidal-to-columnar transformation in *Xenopus* bottle cells, membrane from apical microvilli is endocytosed and trafficked to the basolateral membrane, creating a net movement of membrane from apical to basolateral domains (*Lee and Harland, 2010*).

Because vesicle trafficking is highly active in dynamic epithelia and stable microtubules failed to form in the *Pten* mutant neural plate, we tested whether trafficking was affected by the loss of PTEN. Rab5, a marker of early endosomes, was distributed in an apical-to-basal gradient in the wild type neural plate. In contrast, Rab5+ vesicles were restricted to the most apical domain of the cells in *Pten* mutants (*Figure 7A,B*). Clathrin, a marker for coated endocytic vesicles, was also more apically restricted in *Pten* than in wild-type neural plate cells (*Figure 7C,D*). The normal distribution of Rab5+ and clathrin+ vesicles was restored in *Pten Pdpk1* double mutant neural plates (*Figure 7A–D*).

To test whether the change in vesicle distribution reflected changes in endocytosis or in apical-to-basal trafficking, we cultured E8.0 embryos in presence of transferrin coupled to Alexa-647 and analyzed the localization of transferrin-647 after 8 hrours of culture (*Christ et al., 2012*). Total transferrin-647 uptake was similar in wild-type and *Pten* neural plate cells. However, while transferrin spread along the apical-basal extent of wild-type cells, transferrin accumulated in the apical region in *Pten* cells (*Figure 7E,F*), suggesting that defects in basal trafficking are coupled to the failure of *Pten* mutants to form a pseudostratified columnar neural epithelium. Similar to the other neural plate phenotypes, basal transport of transferrin was rescued in *Pten Pdpk1* double mutants, but was not rescued by treatment of *Pten* with MK-2206 (*Figure 7G,H*).

## Discussion

Mouse embryos that lack PTEN have an unprecedented defect in morphogenesis of the neural tube. In *Pten* mutant embryos, a SOX2+ neural epithelium forms, shows normal segregation of apical and basal markers, is patterned by developmental signals, and proliferates normally. However, the mutant cephalic neural epithelium fails to undergo the transition from a cuboidal to a tall, columnar pseudostratified epithelium; instead, the mutant neural plate is thin, wide and irregularly folded, and cephalic neural tube closure fails completely.

Phosphoinositides have been described as key regulators of apical-basal polarity (*Martin-Belmonte et al., 2007*; *Shewan et al., 2011*), and indeed the *Pten* mutants have a profound defect in the organization of the third (apical-basal) dimension of the neural epithelium. However, the traditional markers of apical-basal polarity are localized correctly in the *Pten* mutant neural plate: Par3, aPKC, ZO1, P-ERM, N-cadherin and F-actin are apically localized, and laminin is basally localized. Based on the enrichment of pAKT in both apical and basolateral membranes of the *Pten* mutant neural plate, apical-basal polarity markers are localized correctly despite high levels of PtdIns(3,4,5)P$_3$ throughout cell membranes.

Despite the important roles of phosphoinositides in mTOR signaling, endocytic sorting, recycling and trafficking (*Di Paolo and De Camilli, 2006*; *Shewan et al., 2011*; *Dibble and Cantley, 2015*), the genetic and chemical genetic data demonstrate that all the phenotypes in the *Pten* neural plate are mediated by increased activity of PDPK1. Although phosphorylated AKT is enriched in all cellular membranes in the mutant neural plate, inhibition of the downstream kinases AKT or mTor does not

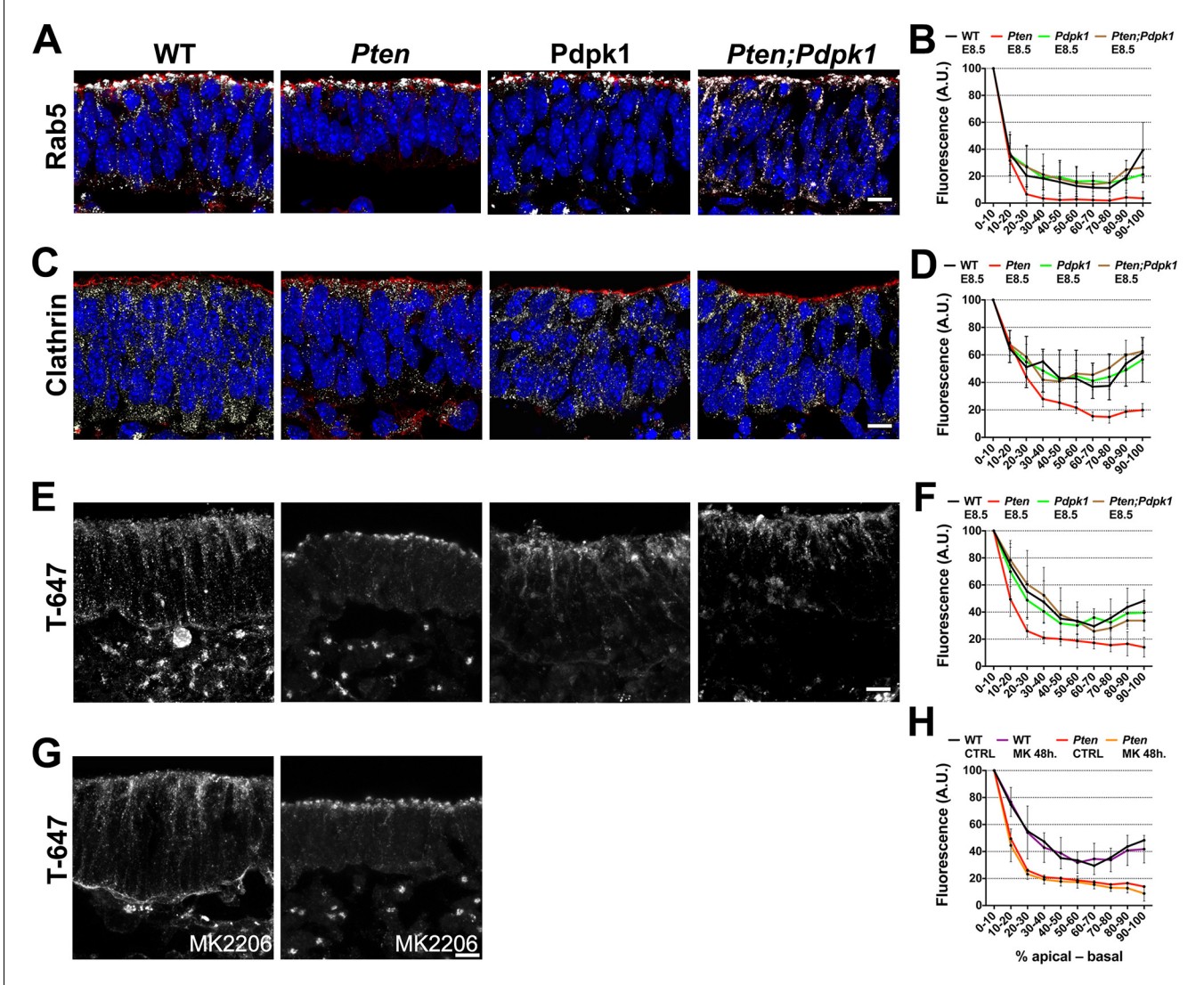

**Figure 7.** Apical-basal trafficking in PI3 kinase pathway mutants. (A– D) Distribution of endosome markers along the apical-basal axis in transverse sections of the cephalic neural plate of E8.5 wild-type, *Pten* △Epi, *Pdpk1* △Epi and *Pten* △Epi *Pdpk1* △Epi embryos. (A) Localization of Rab5, an early endosome marker. (B) Distribution of Rab5 along the apical-basal axis, normalized to a maximum value of 100. (C) Localization of clathrin. (D) Distribution of clathrin along the apical-basal axis, normalized to a maximum value of 100. (E) Uptake of Transferrin-Alexa 647 after 8 hr of embryo culture. Transverse sections of cephalic neural plate of E8.5 wild-type, *Pten* △Epi, *Pdpk1* △Epi and *Pten* △Epi *Pdpk1* △Epi embryos. White signal is the native Alexa 647 fluorescence. (F) Distribution of Alexa-647 signal along the apical-basal axis. Transferrin-647 accumulates apically in the *Pten* △Epi but not in *Pten* △Epi *Pdpk1* △Epi double mutants. (G) Transverse sections of cephalic neural plate of E8.5 wild-type and *Pten* △Epiembryos treated in utero with MK-2206 for 48 hr and then cultured with 50 μg/ml of Transferrin-647 and MK-2206 for 8 hr. (H) Distribution of Alexa-647 along the apical-basal axis is not affected by MK-2206 treatment. Images are Z-projections of 3 optical sections of 1 μm each. Red is phalloidin. Blue is DAPI. Scale bars = 10 μm.

modify the *Pten* mutant phenotype, whereas removal of *Pdpk1* rescues all aspects of the *Pten* phenotype. We therefore conclude that it is the inappropriate PtdIns(3,4,5)P$_3$-stimulated activity of PDPK1, and not changes in levels of other phosphoinositides or in the activity of AKT or mTorc1, that mediates all the morphogenetic defects seen in the *Pten* mutant neural epithelium.

Perhaps the most striking cellular difference between the *Pten* and wild-type neural plate cells is the absence of stable apical-basal microtubule arrays in the mutant. The formation of noncentrosomal apicobasal microtubule arrays, with apical minus-ends and basal plus-ends, is a hallmark of columnar epithelia (*Bré et al., 1987*; *Jaulin and Kreitzer, 2010*). Consistent with a requirement of

microtubule arrays for apical-basal trafficking in columnar epithelia (*Jaulin and Kreitzer, 2010*; *Rodriguez-Boulan and Macara, 2014*), basal trafficking of apically endocytosed transferrin fails in the *Pten* neural plate. Recent work showed that PTEN can bind directly to microtubule-associated vesicles (*Naguib et al., 2015*), suggesting that PTEN could play a direct role in apical-to-basal trafficking in the neural plate. The data show that the PTEN is required for organization of stable arrays of apical-basally oriented microtubules, which may both stabilize the long axis of the cell and promote the redistribution of membrane from the apical to the basolateral domains of neuroepithelial cells, leading to the transition from a cuboidal to a columnar epithelium.

At the same stage (between E8.0 and E8.5) when wild-type neural cells begin to elongate and form arrays of apical-basal stable microtubules, cells of the neural plate are also reorganizing in the plane of the epithelium to become more hexagonally packed. At E8.0, cell packing in both the wild-type and *Pten* mutant anterior neural plate is irregular and includes the rosette-like arrangements that are a hallmark of dynamic epithelia (*Blankenship et al., 2006*). By E8.5, wild-type cells have resolved into a more regular packing pattern and fewer rosettes are observed, while the *Pten* neural plate continues to have many rosette-like cell arrangements.

Pten-dependent, Akt-independent changes in cell packing have also been observed in the *Drosophila* wing disc, where the effect of *Pten* mutations was attributed to a defects in the remodeling of adherens junctions (*Bardet et al., 2013*). Similar to what we observed in the cephalic neural plate of the mouse *Pten* mutant, *Drosophila Pten* mutant wing disc epithelial cells have fewer neighbors than seen in a regular hexagonal array. In the *Drosophila* case, high levels of myosin-II are preferentially seen on short cell edges of *Pten* mutant cells. In contrast, myosin-II is anisotropically distributed in the both the wild-type and mutant E8.0 mouse neural plate, and it can be enriched at either long or short cell edges. The anisotropic distribution of myosin-II persists in the E8.5 *Pten* mutant, while myosin-II becomes enriched at all cell edges in the E8.5 wild-type neural plate, probably in preparation for the next phase of neural tube closure, actomyosin-mediated apical constriction. Thus the loss of PTEN blocks the maturation of cell packing in the neural plate, but there is no simple relationship between the *Pten* phenotype and the distribution of myosin-II.

The abnormal planar cell packing and the absence of apical-basal microtubule arrays in the *Pten* neural plate appear to be coupled: they occur simultaneously and both depend on regulated activity of PDPK1. The coupling of these two phenotypes is consistent with known links between apical junctions and microtubule arrays. Apical adherens junctions are sites for anchorage of noncentrosomal microtubule arrays (*Meng et al., 2008*; *Gavilan et al., 2015*). Microtubules dynamics, in turn, can regulate the stability of adherens junctions (*Meng et al., 2008*; *Waterman-Storer et al., 2000*), supporting the existence of a positive feedback loop that couples stable adherens junctions and microtubule arrays. We propose that a target of PDPK1 in the *Pten* mutant neural plate inhibits stabilization of apical junctions, which, in turn, blocks the formation of the noncentrosomal microtubule arrays required for elongation of cells in the neural plate (*Figure 8*). The direct target of PDPK1 in this process is not known; one possibility is that inappropriate activity of PDPK1 promotes dynamic fluctuations in the activity of aPKC and/or PKN. PtdIns(3,4,5)P$_3$-tethered PDPK1 is sufficient to activate these two classes of kinases (*Balendran et al., 2000*) and aPKC can regulate both apical junctions and microtubule organization (*Harris and Tepass, 2008*; *Harris and Peifer, 2007*).

PTEN has many roles in mammalian brain development, including control of cell size (*Kwon et al., 2001*), neuronal differentiation and migration (*Yue et al., 2005*), synapse structure and synaptic plasticity (*Fraser et al., 2008*; *Sperow et al., 2012*) and axon regeneration (*Park et al., 2008*). Human mutations in one copy of the *PTEN* gene are associated with a variety of abnormalities in brain development, including megalencephaly and focal cortical dysplasia, which can lead to autism and pediatric epilepsy (*Hevner, 2015*; *Jansen, et al., 2015*; *Zhou and Parada, 2012*). Our findings define a profound, very early role of PTEN in the organization of the brain that is likely to contribute to the human syndromes caused by PTEN haploinsufficiency.

PDPK1-dependent changes in epithelial stability could also play an important role in tumors that lack PTEN. Mutations in PI3 kinase pathway are extremely common in tumors: for example, nearly 80% of cases of endometrial carcinoma (non-ultramutated samples) have inactivating mutations in *PTEN* (*Cancer Genome Atlas Research Network et al., 2013*) and 45% of human luminal A breast tumors harbor activating mutations in *PIK3CA* (*Cancer Genome Atlas Network, 2012*). Previous studies provided evidence that anchorage-independence and xenograft growth of breast cancer cells carrying the activated H1047R *PI3KCA* allele depended on PDPK1 but not AKT

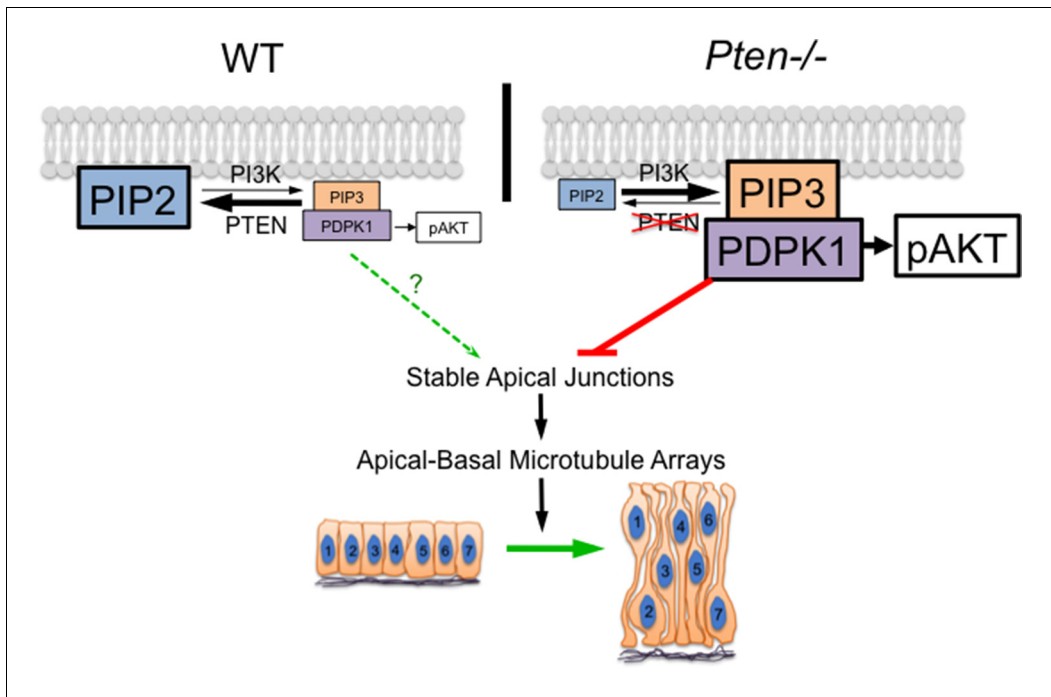

**Figure 8.** A model for the role of PTEN in the formation of the pseudostratified columnar epithelium. PDPK1 is anchored to the plasma membrane by PtdIns(3,4,5)$P_3$ (PIP3), which is made by PI3 kinase (PI3K) and degraded by PTEN. In the *Pten* mutant, increased PIP3 recruits high levels of PDPK1 to the membrane, where it is activated. Activated membrane-associated PDPK1 has two targets: activated PDPK1 generates high levels of pAKT; in a separate pathway, high levels of membrane-associated PDPK1 inhibit the formation of stable apical junctions. Stable apical junctions are required for the formation of stable apical-basal microtubule arrays, which mediate apical-to-basal trafficking in the neural epithelium, allowing elongation and tight packing of cells in the neural epithelium. In WT, PDPK1 is not required for formation of the pseudostratified neural epithelium, although the delay in neural tube closure in *Pdpk1* mutants may reflect a subtle role for the protein in epithelial organization. DOI: 10.7554/eLife.12034.021

(*Gagliardi et al., 2012*) and phosphoproteomic analysis of cell lines with activating *PI3KCA* mutations identified cases in which PDPK1 activity, but not AKT activity, was required for tumorigenicity (*Vasudevan et al., 2009*). The data presented here demonstrate that PtdIns(3,4,5)$P_3$-dependent PDPK1 activity is an important consequence of the absence of PTEN in vivo, even in the absence of activation of AKT. Our findings highlight the importance of identifying the relevant PDPK1 targets during mouse development, in PTEN-associated developmental syndromes, and in tumors.

## Materials and methods

### Mouse strains

The mutant alleles used here have been described previously: *Pten^flox* (*Trotman et al., 2003*), *Pdk1^flox* (MGI designation: *Pdpk1*) (*Lawlor et al., 2002*), *R26-Pik3ca^H1047R* (Jackson Laboratories, Bar Harvor, ME. Stock #016977). The epiblast specific-expressing CRE line is *Sox2-CRE* (*Hayashi et al., 2002*). The Wnt-reporter line used was TOPGAL (*DasGupta and Fuchs 1999*). The genotype of the *Pten* ΔEpi (epiblast-deleted) embryos is *Sox2-Cre/+; Pten^flox/Pten^null*. The genotype of the *Pdpk1* ΔEpi embryos is *Sox2-Cre/+; Pdpk1^flox/Pdpk1^null*. The genotype of the *Pten Pdpk1* ΔEpi double mutants is *Sox2-Cre/+; Pten^flox/Pten^null; Pdpk1^flox/Pdpk1^null*. We generated the *Pten* and *Pdpk1* deleted (*null*) alleles by crossing conditional mice with *Sox2-Cre*, taking advantage of *Sox2* activity in the female germ line. The X-linked *GFP* transgene was a gift from Anna-Katerina Hadjantonakis (*Hadjantonakis et al., 2001*). *Pten* mutants were congenic in CD1, and all other lines, except *R26-Pik3ca^H1047R* (FVB), were backcrossed to CD1 for at least four generations before analysis. For timed pregnancies, noon on the day of the vaginal plug was E0.5.

## In utero embryo drug treatment

Pregnant females were injected intraperitoneally (i.p.) following standard procedures. A final volume of 0.5 ml was injected. Treatments were as follows: 25 mg/kg/day of LY294002 (Selleckchem, Houston, TX) diluted in DMSO at E7.5; 120 mg/kg/day of MK-2206 (from the Baselga Laboratory; commercially available from Selleckchem) diluted in Captisol at E7.5 or E6.5 and E7.5; 3 mg/kg/day of Rapamycin (Sigma, St. Louis, MO) diluted DMSO at E6.5 and E7.5. Embryos were harvested at E8.5.

## Scanning electron microscopy

Embryos for SEM were fixed in 2.5% glutaraldehyde overnight at 4°C, processed using standard procedures and imaged with a Zeiss Supra 25 Field Emission Scanning Electron Microscope.

## LacZ staining and in situ hybridization

β-Galactosidase activity was detected using standard described protocols (*Hogan et al., 1994*). Whole-mount in situ hybridization was performed on embryos following standard methods (*Eggenschwiler and Anderson, 2000*). The *Brachyury* (*Wilkinson et al., 1990*), *En2* (*Joyner and Martin, 1987*), *Krox20* (*Wilkinson et al., 1989*), *EMX2* (*Simeone et al., 1992*), *Fgf8* (*Tanaka et al., 1992*), and *Axin2* (*Jho et al., 2002*) in situ probes were previously described. The embryos were photographed using an HRC Axiocam (Zeiss, Germany) fitted onto a stereomicroscope (Leica, Germany).

## Immunostaining

Embryos were dissected in ice-cold or room temperature PBS/4% BSA and processed for imaging following established protocols (*Lee et al., 2010*). Immunofluorescence staining was performed with Alexa Fluor-conjugated secondary antibodies (Invitrogen, Waltham, MA) diluted 1:400. Sections were counterstained with DAPI (1:2000) to stain nuclei. All images shown are from the cephalic neural plate.

Rhodamine-phalloidin (Invitrogen) was used at 1:200. ARL13b antibody (*Caspary et al., 2007*) was used at 1:2000. Commercial antibodies were: Sigma: γ-tubulin (T-6557), 1:1000 for immunofluorescence (IF); α-Tubulin (T5168) 1:1000 for IF, 1:3000 for western blots (WB); acetylated α-Tubulin (T7451) 1:1000 for IF and 1:3000 for WB. Santa Cruz, Dallas, TX: GAPDH (sc-32233), 1:5000 for WB. Invitrogen: ZO1 (33-9100), 1:200 for IF. Cascade Biosciences, Winchester, MA: Pten (ABM2052), 1:1000 for IF. Cell Signaling, Danvers, MA: Pten (9559) 1:500 for IF; S6 (2217) 1:2000 for WB; pS6 (2211) 1:1000 for WB; pAKT Ser473 (9271) 1:1000 for WB; pAKT Thr308 (2965) 1:1000 for WB; AKT (9272) 1:1000 for WB; Rab5 (3547) 1:100 for IF; Clathrin Heavy Chain (4796) 1:100 for IF; pMLC2 (3671), 1:1000 for WB; acetylated α-Tubulin (5335) 1:3000 for WB. Hybridoma Bank, Iowa City, IA: Nkx2.2 (74.5A5) 1:100 for IF; Nkx6.1 (F55A10) 1:50 for IF. Covance, Princeton, NJ: MHCIIB (CMII-23; PRB-445P), 1:50 for IF, and 1:1000 for WB. Abcam, Cambridge, MA: FOXA2 (AB40874) 1:800 for IF. Millipore, Billerica, MA: Olig2 (AB9610) 1:200 for IF; SOX2 (AB5603) 1:1000 for IF.

For immunofluorescence, samples were mounted using Vectashield (Vector Labs, Burlingame, CA) or ProLong Gold (Life Technologies, Carlsbad, CA) mounting media, and slides were imaged with SP5 and SP8 confocal microscopes (Leica) with a 63 × 0.5 NA lens, at a resolution of 1024 × 1024. In transverse sections, maximum intensity was set in the apical domain, and images with apical non-saturated signal on the neural plate were taken. *En face* images are Z-projections of 3–5 single optical sections taken every 0.3 μm. Images were analyzed using Volocity software (PerkinElmer, Waltham, MA). The immunofluorescence data presented in the figures are representative images of at least three embryos.

## Fluorescence signal quantification

Pixel intensity along the apicobasal axis of the neural plate was determined on Z-stack projections of 5 optical sections taken every 1 μm (grayscale). Pixel intensity values were taken from lines 20 pixels wide traced with ImageJ. Graphical distribution of pixel intensity average (n≥3 embryos) was generated using Prism6 with normalized values.

## Transferrin uptake assay

E8.0 embryos with intact yolk sac and ectoplacental cone were dissected in 37°C DMEM/F12 containing 10% FBS. After dissection, 5 embryos were transferred to a glass bottle (Roller Bottle System) containing 5 ml of 50% rat serum/50% DMEM/F12 and incubated at 37°C with 5% $CO_2$ and 10% $O_2$. Transferrin-Alexa 674 (Molecular Probes, Eugene, OR. #Ta3366) was diluted in the culture media to 50 µg/ml, as described (*Christ et al., 2012*). After 8 hr, the yolk sac was removed and the embryos were fixed in 4% PFA for 2 hr at 4°C and mounted for cryosectioning following established protocols (*Lee et al., 2010*). Images were taken from transverse sections of the cephalic region using a SP5 Leica confocal microscope collecting the native signal from Transferrin-Alexa 674.

## Morphometric analysis

Neural plate height of cephalic region was measured following a previously described method (*Grego-Bessa et al., 2015*). Apical surface area quantification of cephalic neuroepithelial cells was determined from *en face* images taken with a Leica SP5 inverted confocal microscope and 63 × 0.5 NA lens, and analyzed by Volocity software (>100 measurements per embryo, n≥3 embryos). For all analyses, n≥3 embryos. Measurements are average ± s.d. Comparisons were made by standard t-test. Prism6 was used for statistical analysis.

For analysis of cell packing, ZO-1, β-Catenin and Phalloidin-Rhodamine staining delineated the apical domain of cephalic neuroepithelial cells. *En face* images of the cephalic region were taken by confocal microscope at 63× of magnification. For two-dimensional cell patterns, the number of edges/cell and the number of vertices formed by 3–7 cells were quantitated manually from at least 3 embryos per genotype (>200 cell vertexes). Data analysis was performed with Excel and Prism6.

## Immunoblotting

A pool of three E8.5 embryos, after removal of the heart, was lysed in Cell Lysis Buffer (Cytoskeleton, Denver, CO. GL36) plus Complete Protease Inhibitor Cocktail (Roche, Germany). Western blots were performed according to standard protocols, and protein was detected with HRP-conjugated secondary antibodies and ECL detection reagents (Amersham, UK).

## Acknowledgements

We thank Dr. Anna-Katerina Hadjantonakis for X-linked GFP transgenic mice and Dr. Dario Alessi for the *Pdpk1* conditional mice. We thank Dr. Jennifer Zallen, Dr. Hadjantonakis and members of Anderson laboratory for helpful conversations and thoughtful comments on the manuscript, Vitaly Boiko and the MSKCC Molecular Cytology Core Facility for valuable technical support. Monoclonal antibodies were obtained from the Developmental Studies Hybridoma Bank, created by the NICHD of the NIH and maintained at The University of Iowa, Department of Biology, Iowa City, IA 52242. The work was supported by R37 HD03455 and R01 NS044385 to KVA, the MSKCC Cancer Center Support Grant (P30 CA008748), and a Beatriu de Pinós postdoctoral Fellowship from Generalitat de Catalunya to JGB.

## Additional information

### Funding

| Funder | Grant reference number | Author |
|---|---|---|
| National Institutes of Health | HD03455 | Kathryn V Anderson |
| National Institutes of Health | NS044385 | Kathryn V Anderson |
| National Institutes of Health | P30 CA008748 | José Baselga<br>Kathryn V Anderson |

The funders had no role in study design, data collection and interpretation, or the decision to submit the work for publication.

## Author contributions
JGB, Conception and design, Acquisition of data, Analysis and interpretation of data, Drafting or revising the article; JB, Conception and design, Acquisition of data, Analysis and interpretation of data; PC, JoB, Conception and design, Contributed unpublished essential data or reagents; TO, Conception and design, Analysis and interpretation of data, Contributed unpublished essential data or reagents; KVA, Conception and design, Analysis and interpretation of data, Drafting or revising the article

## Ethics
Animal experimentation: This study was performed in strict accordance with the recommendations in the Guide for the Care and Use of Laboratory Animals of the National Institutes of Health. All of the animals were handled according to approved Institutional Animal Care and Use Committee (IACUC) protocol (02-06-013) of Memorial Sloan Kettering Cancer Center

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
