## [Decision Letter]

Thank you for submitting your work entitled "PTEN and PDK1 regulate formation of the columnar neural epithelium" for consideration by *eLife*. Your article has been reviewed by two peer reviewers, one of whom, Joseph Gleeson, is a member of our Board of Reviewing Editors, and the evaluation has been overseen by the Reviewing Editor and a Senior Editor.

The reviewers have discussed the reviews with one another and the Reviewing editor has drafted this decision to help you prepare a revised submission.

Summary:

In this elegant paper the authors use a combination of genetic and pharmacological tools to elucidate the mechanism underlying the failure to form a pseudostratified neuroepithelium in *Pten* conditional mutant mice. They demonstrate that loss of *Pten* and the resulting hyperactivation of PDK1 significantly disrupt this morphogenetic process, causing neural cells to remain cuboid instead of elongated, to lose their apical microtubule arrays and the ability to traffic endocytic vesicles. The linearity of the pathway for this phenotype was verified by rescue of the phenotype by conditional removal of Pdk1, a downstream effector, whereas another *Pten* phenotype (cardia bifida) was not rescued. They further demonstrate that, perhaps surprisingly, this pseudostratified phenotype is not dependent on elevated downstream AKT or mTORC1 activity, suggesting that an unknown mechanism downstream of PDK1 is involved. Elevated PDK1 signaling thus likely causes abnormal cell shape, apical junctions and endocytic vesicle trafficking defects in *Pten*-related neurodevelopmental disorders.

The paper is well written and well executed, with figures fully supporting the conclusions. One of the major strengths is the phenocopy with the gain of function *Pik3ca* mutation, and the rescue with the *Pdk1* mutant.

Essential revisions:

1) The authors show that PI3K and PDK1 are *not* required for the normal formation of the pseudostratified neuroepithelium. Thus, the diagram proposed in Figure 8 is a bit misleading, as it suggests that PI3K-PDK1 signaling promotes the formation of stable apical junctions, which the authors then argue are causally linked to the formation of the pseudostratified neuroepithelium. The various phosphoinositides regulate multiple aspects of TOR signaling, endocytic sorting, recycling and trafficking. A more critical discussion of the proposed role of PI3K-PDK1 signaling in normal and abnormal morphogenesis is warranted to explain these findings. While the genetic investigations offer strong support, researchers in the phosphoinositide field might question the link to endosomal trafficking and microtubule post-translational modification.

2) Even though the study focuses on neural plate defects in *Pten* mutant mice there is little mention of *Pten*- or PI3K-associated neurodevelopmental disorders, and more emphasis is placed on cancer in other organs. It would be appropriate to incorporate in the Discussion recent findings on the role of excessive PI3K signaling in brain overgrowth syndromes and cortical malformations (see for example recent reviews by Jansen et al, Brain 138:1613 2015, and Hevner, Sem Perinatol 39: 36 2015).

3) Figure 2 could be improved. Please provide better-matched images and more detailed timing, if necessary, to show time course in more detail.

4) There are many other anatomical locations with pseudostratified epithelium, raising the question if *Pten* is required for their formation.

---

## [Author Response]

*1) The authors show that PI3K and PDK1 are* not *required for the normal formation of the pseudostratified neuroepithelium. Thus, the diagram proposed in Figure 8 is a bit misleading, as it suggests that PI3K-PDK1 signaling promotes the formation of stable apical junctions, which the authors then argue are causally linked to the formation of the pseudostratified neuroepithelium. The various phosphoinositides regulate multiple aspects of TOR signaling, endocytic sorting, recycling and trafficking. A more critical discussion of the proposed role of PI3K-PDK1 signaling in normal and abnormal morphogenesis is warranted to explain these findings. While the genetic investigations offer strong support, researchers in the phosphoinositide field might question the link to endosomal trafficking and microtubule post-translational modification.*

The reviewers are absolutely correct that the data indicate that PI3K and PDK1 are not required for formation of the pseudostratified neuroepithelium. We have revised both the text and Figure 8 to better illustrate that our results show that the *Pten/Pik3ca* phenotype is the result elevated or ectopic activity of PDK1, and do not define the normal role of PDK1 in this process.

We now refer more directly to the other activities of phosphoinositides that might be relevant for the phenotypes we observe. We point out that the genetic experiments argue that, remarkably enough, it is PDK1 activity that is responsible for the observed phenotypes, rather than mTor activity or other PDK1-independent effects of phosphoinositides. We appreciate that the links to endosomal trafficking and microtubule post-translational modification are novel and will require further study, but the data clearly implicate PDK1 in these processes.

2) Even though the study focuses on neural plate defects in Pten mutant mice there is little mention of Pten- or PI3K-associated neurodevelopmental disorders, and more emphasis is placed on cancer in other organs. It would be appropriate to incorporate in the Discussion recent findings on the role of excessive PI3K signaling in brain overgrowth syndromes and cortical malformations (see for example recent reviews by Jansen et al, Brain 138:1613 2015, and Hevner, Sem Perinatol 39: 36 2015).

This is also an excellent point. The Discussion has been modified to better highlight the implications of this work for neurodevelopmental disorders.

3) Figure 2 could be improved. Please provide better-matched images and more detailed timing, if necessary, to show time course in more detail.

The panels in Figure 2 are somite-matched for staging, which we indicate in the revised Figure. We have included additional panels in Figure 2 to show the time course of tubulin acetylation in the wild-type and mutant neural plate, and we now describe the acetylation of the microtubules in the floor plate. We have also added new panels to Figure 2—figure supplement 1 to show both the time course of tubulin acetylation and the partial colocalization of PTEN protein with acetylated tubulin at E8.5 and E9.5.

4) There are many other anatomical locations with pseudostratified epithelium, raising the question if Pten is required for their formation.

This is an interesting point. The epiblast-specific deletion of *Pten* that we have studied causes developmental arrest at approximately E9.0. We were therefore not able to examine other pseudostratified epithelia; analysis of additional pseudostratified epithelia using other Cre lines is interesting topic for the future.